# Physical activity and exercise recommendations for people receiving dialysis: A scoping review

Kelly Lambert[1]*, Courtney J. Lightfoot[2], Dev K. Jegatheesan[3], Iwona Gabrys[4], Paul N. Bennett[5]

1 School of Medicine, Faculty of Science, Medicine and Health, University of Wollongong, and Illawarra Health and Medical Research Institute, Wollongong, New South Wales, Australia, 2 Department of Health Sciences, Leicester Kidney Lifestyle Team, University of Leicester, Leicester, United Kingdom, 3 Department of Nephrology, Princess Alexandra Hospital, Woolloongabba, Queensland, Australia, 4 Alberta Kidney Care North, Alberta Health Services, Edmonton, Alberta, Canada, 5 Clinical Health Sciences, University of South Australia, Adelaide, Australia and Satellite Healthcare, San Jose, California, United States of America

* klambert@uow.edu.au

**Data Availability Statement:** The data underlying the results presented in the study are available from the Open Science Framework https://osf.io/g6x8b/.

## Abstract

### Introduction

Remaining physically active is important to patients undertaking dialysis, however, clinical recommendations regarding exercise type, timing, intensity, and safety precautions vary. The purpose of this scoping review was to analyse and summarise recommendations for physical activity and exercise for people undertaking dialysis and identify areas that require further research or clarification.

### Materials and methods

A scoping review of literature from five bibliographic databases (Medline, Scopus, Web of Science, CINAHL, and SPORTDiscus) was conducted. Eligible articles included consensus guidelines, position statements, reviews, or clinical practice guidelines that included specific physical activity and exercise recommendations for people undertaking dialysis. Key search terms included "kidney disease" OR "kidney failure" OR "chronic kidney disease" OR "end stage kidney disease" *AND* guideline* OR consensus OR "position statement" OR prescription OR statement *AND* exercise OR "physical activity". Hand searching for relevant articles in all first twenty quartile 1 journals listed on SCImago under 'medicine—nephrology' and 'physical therapy, sports therapy and rehabilitation' using the terms 'exercise and dialysis' was undertaken. Finally, home pages of key societies and professional organisations in the field of sports medicine and nephrology were searched.

### Results

The systematic search strategy identified 19 articles met the inclusion criteria. Two were specific to pediatric dialysis and three to peritoneal dialysis. Whilst many publications provided recommendations on aerobic exercise, progressive resistance training and flexibility, few provided explicit guidance. Recommendations for the intensity, duration and frequency

**Funding:** KL. Funding was received from the Illawarra Health and Medical Research Institute for this work. https://www.ihmri.org.au/ The funders had no role in study design, data collection and analysis, decision to publish, or preparation of the manuscript.

**Competing interests:** The authors have declared that no competing interests exist.

of aerobic and resistance training varied. Discrepancies or gaps in guidance about precautions, contraindications, termination criteria, progression, and access site precautions were also apparent.

## Conclusion

Future guidelines should include specific guidance regarding physical activity, safety precautions, and timing and intensity of exercise for individuals who undertake dialysis. Collaborative multidisciplinary guideline development and appropriate exercise counselling may lead to increased participation in physical activity and exercise and facilitate better patient outcomes.

## Introduction

Globally, over seven million people with kidney failure require kidney replacement therapy to maintain life [1]. The most common treatment option for kidney failure is dialysis, with its incidence increasing by 43% since 1990 [2]. In the dialysis population, the lack of physical activity and exercise is associated with poorer quality of life [3], lower physical functioning [4], greater bodily pain [5], increased hospitalisation, and overall poorer survival [5].

Remaining physically active is important to patients undertaking dialysis [6–9]. Exercise also produces many benefits including improved physical function [3], muscle mass and strength [10]. However, there are numerous barriers to undertaking regular physical activity and exercise. These may be physiological [11], physical [12], psychological, practical [13], as well as structural (such as family support or lack of access to exercise professionals or programs [13, 14]). Unfortunately, access to exercise professionals and maintaining the sustainability of structured dialysis exercise programs has been an ongoing challenge for dialysis centers [14]. Dialysis clinicians who aim to support patient physical activity are restricted due to variation and limited guideline recommendations [15].

We sought to undertake a scoping review of the literature regarding guidance for physical activity and exercise for people with kidney failure undertaking dialysis. The specific objectives of this review were to:

1. Analyse and summarise recommendations for physical activity and exercise for people undertaking dialysis.

2. Identify areas that require further research or clarification.

## Materials and method

### Rationale

Scoping reviews are intended to explore the sources and types of evidence available, and systematically characterise the existing literature on a broad scale. We followed the five stages of a scoping review outlined by previous authors [16, 17]: (i) identifying the research questions; (ii) identifying relevant literature; (iii) selecting evidence; (iv) collating evidence; and (v) synthesising evidence. This scoping review is reported according to the Preferred Reporting Items for Systematic Reviews and Meta-Analyses (PRISMA) Scoping Review guidelines [18]. The

protocol was published on the OpenScience Framework on 11 September 2020 (Registered form osf.io/g6x8b, Registration DOI: 10.17605/OSF.IO/TNRS5).

## Eligibility criteria

Articles considered eligible for inclusion were consensus guidelines, position statements, reviews, or clinical practice guidelines that included specific physical activity and exercise recommendations for people undertaking dialysis. No age restriction was applied, thus documents for both adults and children were eligible for inclusion. Database searches were restricted to English only. However, documents in the grey literature were excluded only if they were unable to be translated by members of the research team. Details of eligibility criteria are included in Table 1.

## Search strategies

Comprehensive searching of five bibliographic databases (Medline, Scopus, Web of Science, CINAHL and SPORTDiscus) was undertaken on September 3rd, 2020 with weekly literature surveillance alerts received until 26 April 2021. All database searches included articles from database inception to search date. The search strategy was informed by similar work in the field of diabetes [19] and guidance from research librarians to refine the search strategy. The key search terms entered included "kidney disease" OR "kidney failure" OR "chronic kidney disease" OR "end stage kidney disease" *AND* guideline* OR consensus OR "position statement" OR prescription OR statement *AND* exercise OR "physical activity". Records from these searches were retrieved and downloaded into Endnote (Thomson Reuters; Version X9, 2018). All citations were then uploaded to the Sys.Rev.com open access document review platform (https://sysrev.com/) to facilitate duplicate screening of abstracts by the research team. One reviewer (KL) examined the full text of all citations. Two reviewers (IG and CL) each reviewed 50% of citations. Disagreements were resolved by discussion amongst the reviewers. For studies that passed initial screening, full text articles were retrieved and reviewed by three members of the team to determine inclusion (KL, IG, CL).

Grey literature searching involved four search strategies: First, the database CINAHL was specifically utilised to capture grey literature. Secondly, the home page search function of the first twenty quartile 1 journals listed on SCImago under 'medicine—nephrology' and 'physical therapy, sports therapy and rehabilitation' were searched using the terms 'exercise and dialysis'. A request for documentation was placed to all members of the Global Renal Exercise Network (GREX) on 15 September 2020. Finally, home pages of key societies and professional organisations in the field of sports medicine and nephrology were searched until 26 April

**Table 1. Scoping review eligibility criteria.**

| Inclusion criteria | Exclusion criteria |
|---|---|
| Published by government or non-government organisations, societies or experts at the national or international level | Abstracts or conference proceedings. Individual studies reporting on the outcomes of interventions regarding exercise in people undertaking dialysis |
| The document must include recommendations or advice regarding physical activity and / or exercise for any person undertaking dialysis | Systematic reviews with no reference to recommendations for exercise |
| Most current version of the document is available | Draft versions |
| Database searches: English only | Grey literature searches: documents unable to be translated into English by the research team |
| Grey literature search: English, Polish and Spanish | |

2021. A complete list of these are shown in S1 Table. The reference lists of documents retrieved were also searched.

## Data extraction

One reviewer (KL) independently extracted information from the eligible articles for data synthesis. A second reviewer (IG) checked the accuracy of data extracted from eligible articles. Extracted data included: organisation name or author; type of exercise recommended; mode of exercise; duration; intensity; frequency; safety considerations or additional commentary specific to people undertaking dialysis.

In order to analyse recommendations for physical activity and exercise, the following definitions were used: *Physical activity* refers to bodily movement of skeletal muscles in any manner that results in energy expenditure [20]. *Exercise* is planned, structured, repetitive body movement intended to produce a health benefit or improve fitness [21].

## Results

The systematic search strategy identified 1451 records from database searches and an additional 24 records from the grey literature. After exclusion of duplicates and application of the screening criteria, 19 publications were eligible for inclusion in the review (Fig 1).

### Characteristics of selected studies

Eight publications (42%) were obtained from the grey literature [22–29]. Thirteen publications were published on behalf of professional associations or foundations (The American College of Sports Medicine [ACSM] [30], Exercise and Sports Science Australia [31]; European Federation of Sports Medicine Association [27]; the Italian Society of Nephrology [32]; the United Kingdom Renal Association [33]; Chilean Society of Nephrology [28, 29]; Amicus Renis Foundation on behalf of the Polish Society of Nephrology [24, 25]; the Renal Foundation of Inigo Alvarez de Toledo [23]; Spanish Society of Nephrology [22]; the Kidney Disease Outcomes Quality Initiative [34] and the Life Options Rehabilitation Advisory Council [26]). Two publications were specific recommendations for pediatric dialysis populations [35, 36]. Three publications were specific to adults undertaking peritoneal dialysis (PD) [24, 28, 37]. Six publications were specific to hemodialysis [22, 23, 26, 29, 32, 33] and six were applicable to both types of dialysis [25, 26, 28, 31, 38, 39]. Four did not specify the type of dialysis. Fourteen of the 19 publications were published in the last decade.

### Recommendations regarding physical activity

Table 2 summarises the recommendations obtained regarding physical activity. A more detailed summary of recommendations obtained is contained in S2 Table. No publication provided a definition of physical activity nor detail on the type of activities considered as physical activity. Few publications also recommended physical activity explicitly. Advice about physical activity ranged from brief advice such as 'perform physical activity' [28, 29] to more explicit information such as 'undertake recreational activity that will contribute to the improvement of cardiovascular and respiratory efficiency' [24, 25]. Recommendations for timing and duration of physical activity were contained in only two publications. One publication provided advice for physical activity based on the level of deconditioning, and commenced at 30–60 minutes per day of walking for those who are highly deconditioned or do not exercise [39]. A second publication recommended 30 minutes or more of continuous physical activity 5 days per week [30].

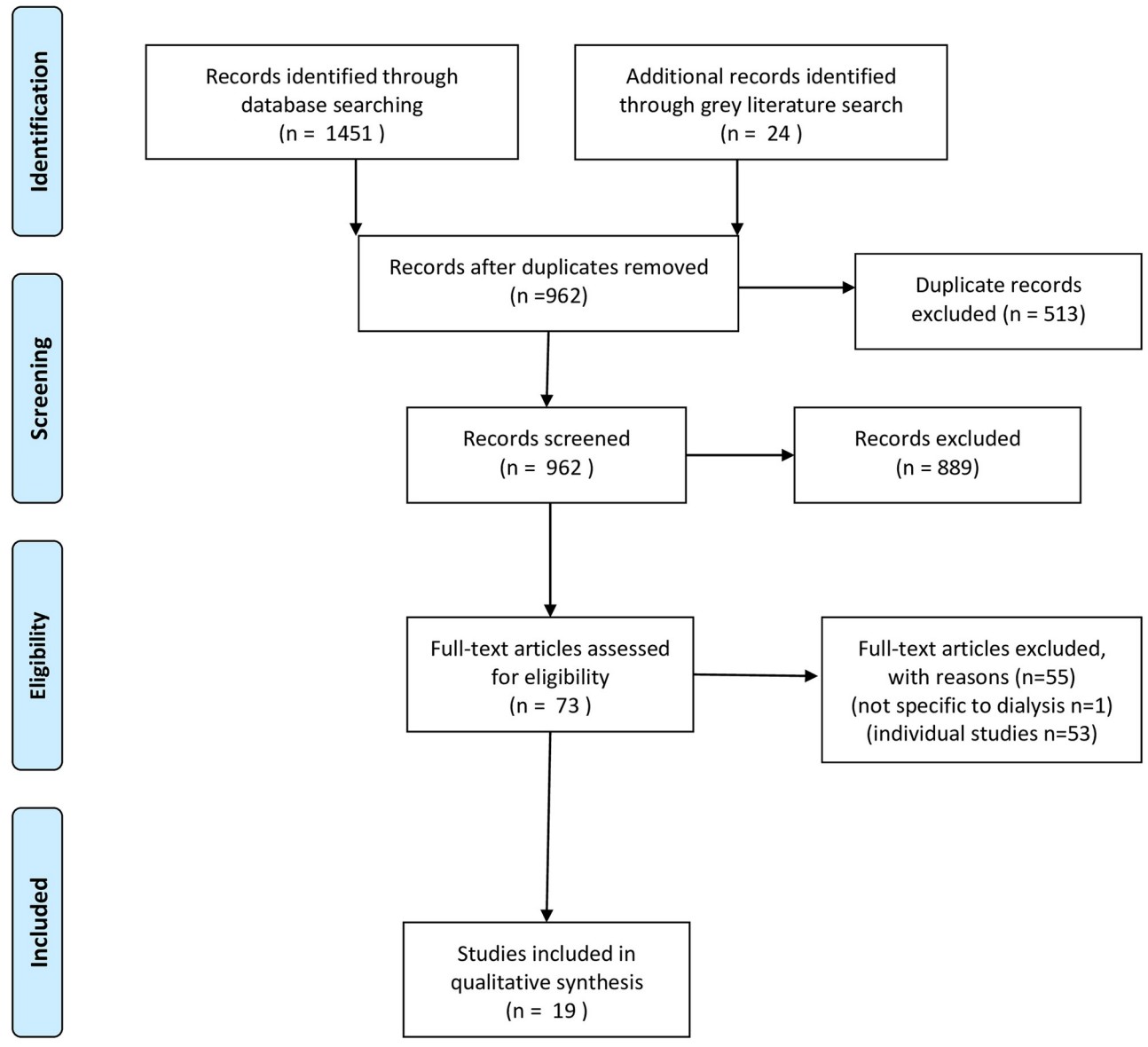

**Fig 1. PRISMA flow chart.** *From*: Moher D, Liberati A, Tetzlaff J, Altman DG, The PRISMA Group (2009). *Preferred Reporting Items for Systematic Reviews and Meta-Analyses: The PRISMA Statement. PLoS Med 6(7): e1000097. doi:10.1371/journal.pmed1000097 **For more information, visit** www.prisma-statement.org.

## Recommendations regarding aerobic exercise

The majority of publications recommended patients undertake aerobic exercise, and S3 Table highlights that ten of the 19 publications made a specific recommendation to include aerobic exercise for people undertaking dialysis [22, 28–33, 35, 36, 39]. Specific examples of the type and mode of aerobic exercise commonly recommended were walking, running, swimming, and cycling (Table 2). No consensus existed between publications about the recommended duration of aerobic exercise, and recommendations varied from 20–60 minutes per day [24–27, 30, 35, 36] to 15–45 minutes per hemodialysis session [23, 31] to 180 minutes per week [31].

**Table 2. Summary of recommendations about physical activity and exercise obtained from papers retrieved in scoping review.**

| Physical activity | |
|---|---|
| Recommendation | **Physical activity is encouraged** [24–26, 28, 29, 34] |
| Examples | Walking, cycling [24, 25] |
| Timing | No detail on ideal timing |
| Duration | **30 minutes per day (or 20 minutes per day if vigorous)** [30, 39] |
| Intensity | **Moderate or vigorous** [30] or measure via step count—minimum 3000 steps per day [39] |
| Frequency | **3–5 days per week** [30, 39] |
| Progression | **Individualised by professional eg exercise physiologist, increase by 3–5 minutes per week until 30 mins per day achieved then increase intensity [30]** |
| Exercise | |
| Recommendation | **Aerobic and resistance training are beneficial** [28–31, 24–26, 35, 36, 39]. Flexibility exercises are also beneficial [26, 27, 31, 39] and for those at risk of falls, balance exercise are advised [27, 39] |
| Type and mode | Aerobic: **prolonged activities using large muscles** eg **walking or cycling** [22, 24–31, 35, 36, 39, 38, 41], swimming [26, 37, 39], dance or jog [27–29, 38], skating, Nordic walking [27], cross country skiing [38], jumping jacks, skipping rope or stationary bike [37] |
| | Intradialytic: **pedalling** [22, 23, 32, 33] or **step device** [30], seated cycle or arm or leg ergometer [27, 31, 32], rolling balls [32] |
| | Resistance: **functional training** [23, 27, 33, 36], weight bearing activity [26, 39], **TheraBand** [26, 33, 35, 36], **weight cuffs** [33], weight machines [26, 35, 36], dumbells [22, 24–6, 31, 35], **free weights** [22, 24–26, 33, 35–37] |
| | Flexibility: **static or proprioceptive neuromuscular facilitation** [30], stretch arms if in wheelchair [22], trunk twists, lateral bends and standing elbow to knee for intermediate and high functioning patients [37], general stretching [26] |
| | Balance: balance exercise to reduce risk of falls [31, 38] |
| Timing | General: Start in 10–15 minute bouts [37, 38], **intradialytic during first 2 hours of HD** [31–33] |
| | Aerobic: can be prior to dialysis or **non dialysis days** [30, 38] |
| | Resistance: can be completed before or during [31] or on the non dialysis day [31]. Complete up to 15 reps or to fatigue [31, 37] |
| | Flexibility: No ideal time described |
| Duration | Aerobic: sessions range from **20–60 minutes** [22, 24–41] |
| | Resistance: **1 set 10–15 reps** [23–26, 30, 31, 35, 36, 39] |
| | Flexibility: 60 seconds per joint for static exercises [31] and 10 minutes per session [31, 39] |
| Intensity | Aerobic: **moderate** [23–27, 30–36, 38, 39] or moderate to vigorous [26, 27] |
| | Resistance: **60–75% 1 RM** [30, 31, 35, 36, 39], or low intensity [37] |
| | Flexibility: **point of tightness or slight discomfort** [30] |
| Frequency | General recommendation: **2–5 x week** [23–30, 32, 34, 38, 40, 41] |
| | Aerobic: **3–5 week** [30, 32, 35, 36, 39] |
| | Intradialytic aerobic exercise 2 x per week [23] **or all sessions** [33] |
| | Resistance: **2–3 x per week** [30, 31, 35, 36, 39] with 48 hours break between sessions [36]. Intradialytic strength exercises 3 x per week [23] |
| | Flexibility: **2–3 x per week** [30] or 5–7 x per week [31, 39] |
| Progression | **Gradual progression over time based on individual tolerance [30, 31, 33]. Increase by 3–5 minutes per week until 30 mins per day achieved then increase intensity [26, 30]**. Progress in periodized blocks of 2–5 weeks [36] |
| Safety and other recommendations | |
| Clearance recommendations | **Medical clearance required before commencement** [24–26, 30–32, 34–36, 38, 39]. |
| | Assessment by sports medicine physician or cardiologist [32] or nephrologist [23–25, 36], diabetes specialist, or specialist physiotherapist [24, 25] |
| | Assessment performed on a mid-week non dialysis day [39] |
| Contraindications | **New within 3 months to hemodialysis, hypo or hyperglycaemia, high prevalent BP >160/100mm/Hg, SOB, peripheral oedema, suspected or recent pericarditis, myocarditis, dissecting aneurysm, DVT, not stable on HD (eg intradialytic hypotension), recent MI, unstable ischemia, uncontrolled arrhythmias, severe pulmonary hypertension, angina, severe headache, dizziness, severe and symptomatic aortic stenosis, uncontrolled hypertension >180/110, excessive IDWG, decompensated heart failure**, ECG changes, **pulmonary embolism**, large pericardial infusions, severe valvular disease, retinal haemorrhage, **fever** > 38.3 degrees, severe peripheral or cardiac neuropathy, severe CKD MBD, **recent stroke or TIA**, physical or cognitive abilities that limit ability to cooperate, **significant anaemia, hyperthyroidism**. [26, 30, 31, 33, 35, 38, 39] |
| | PD related: peritonitis, catheter site infection, non-functioning catheter [24, 25] |

(*Continued*)

**Table 2.** (Continued)

| | |
|---|---|
| Precautions to exercise | **Use RPE not HR as indicator of exercise intensity** [30, 35, 36] |
| | **Exercise after HD may increase risk of hypotension** [30] |
| | **Intradialytic exercise in first half of dialysis** [30, 35, 36] |
| | regular BP and ECG in those with history of abnormal serum potassium or unstable potassium [31, 35] |
| | Closely supervise those with hx of diabetic nephropathy or cardio- renal syndrome [31] |
| | Avoid exercise in hot environments including saunas [38, 39] |
| | **Monitor BGL**. In those with diabetes especially if exercise is longer than 1 hour **or in the case of hypoglycemia, then consume carbohydrate**. Exercise with a partner if on insulin. [23–26, 33, 39] |
| | Terminate **if BP > 220/105 mm Hg** [33] **or symptomatic or discomfort / pain** [23, 26, 33, 39] |
| | Avoid commencement if not adequately dialysed or fluid overloaded [26] |
| | **Prolonged cool down is recommended** [24, 25, 33, 35, 37, 39] |
| Fistula arm advice | **Do not exercise fistula arm during dialysis** [30, 39] |
| | Avoid upper limb exercise while temporary catheter in-situ [31, 39] |
| | **Avoid placing weight on fistula site** [30, 35, 36] |
| PD specific advice | Avoid exercises that involve twisting of the torso [38], avoid forced breathing / Valsalva manoeuvre [23–25, 38], exercise while lying on stomach or static abdominal exercises [24, 25, 38], avoid increasing intraabdominal pressure [32, 35] |
| | Swimming: clean access site after swimming [37, 26], dress catheter site if swimming [35], avoid swimming in contaminated water [22] |
| | Fluid in cavity: Patients should attempt to exercise while full with fluid [24, 25, 35–37] except if hernia, leak or hypotension during exercise [24, 25]. Conduct exercise while partially full [23, 26, 38, 39], or dry to avoid discomfort [26, 27, 31, 37] or to achieve higher level of exertion [39] |
| Other safety comments | **Exercise should be individualised** [23–26, 30, 36, 38, 39] |
| | Exercise while supervised by professional [32, 33] |
| | Patients with neuropathy should be educated about safe foot care [26] |

**Bolded** statements Bolded statements are derived from recommendations using systematic reviews or randomised controlled trials (higher quality evidence) rather than opinion only.

Abbreviations: BGL blood glucose level; BP blood pressure; DVT deep vein thrombosis; ECG: Electrocardiogram; HD hemodialysis; HR heart rate; IDWG interdialytic weight gain; MI myocardial infraction; PD: Peritoneal dialysis; RM repetition maximum; RPE: Rating or Perceived Exertion; SOB shortness of breath; TIA transient ischemic attack.

There was significant variation in the level of intensity suggested for aerobic exercise across publications. Ten publications suggested intensity should be of the moderate level (i.e. classified as 40–60% $VO_2$ max [26, 30, 32, 35, 36] or 10–12 [23] / 11–13 [30, 31, 35, 36] / 12–13 [30] / 12–16 [26]/ or 11–16 [24, 25, 39] on the Borg scale of rating of perceived exertion [RPE]; S3 Table). One publication [33] suggested moderate to vigorous intensity (40–75% $VO_2$ max or 12–15 on the Borg scale). Another suggested moderate (40–60% $VO_2$ max or 11–13 on the Borg scale) or vigorous intensity [27] (60–80% $VO_2$ max or 12–18 for vigorous on the Borg scale). Others suggested light to moderate intensity (30–40% increase from resting heart rate or 11–14 on the Borg scale [24, 25]). Six publications made no specific recommendation regarding intensity of aerobic exercise [22, 28, 29, 37, 38, 40].

In general aerobic exercise was recommended 3–5 times per week (Table 2). However, these recommendations did vary. For example, six publications provided broad recommendations for frequency ranging from daily [27, 38] to as often as possible [24, 25]. Seven provided more specific recommendations including 1–2 times per week [39]; twice per week on non-dialysis days [32]; 3 times per week during dialysis [23, 33]; 3 to 5 times per week [26, 35]; 3 times per week [40]; or most days [34, 36]. Recommendations regarding frequency were not provided in six publications [22, 28–31, 37].

## Recommendations regarding resistance exercise

Most publications recommended resistance training for dialysis patients [22–27, 30, 31, 33, 35–37, 39]. One publication made a specific recommendation to include resistance training for the arteriovenous fistula (AVF) arm only [22]. Six publications made no recommendation regarding resistance training [24, 25, 28, 29, 34, 38].

Table 2 provides detail on the type of resistance exercises recommended. The most common type of resistance exercises recommended were weight machines, dumbells and free weights [22–26, 31, 33, 35–37] or resistance bands [23, 26, 31, 33, 35, 36]. Another publication recommended multi-joint weight-bearing exercises but did not provide specific details on the types of exercises [39]. Recommended resistance exercises were categorised by one publication according to level of functioning [37].

Recommendations regarding the timing, duration, and intensity of resistance exercises were variable. The most frequently reported number of repetitions was one set of 10–15 repetitions in the range of 60–75% of 1 repetition maximum (RM) (Table 2) [23–26, 30, 31, 35, 36, 39]. Several other publications recommended a lower number of repetitions (5–10 repetitions [24, 25, 27, 37]) (S3 Table). The rationale for the reduced number of repetitions was highly variable. One publication indicated a lower number of repetitions (e.g. 1–2 sets of 6–10 repetitions) for low functioning individuals, and for high functioning individuals 2–3 sets of 12–15 repetitions were recommended [37]; however, they did not provide detail on intensity level [37]. One publication provided specific details on intensity on dialysis (1 set to fatigue or 12–15 repetitions) and non-dialysis days (1 set to fatigue or 10–15 repetitions) as well as intradialytic intensity (1 set to fatigue or 12–15 repetitions) [31]. One publication provided detail on intensity but not number of repetitions or sets [34]. Another provided details on number of repetitions and sets but not intensity [23]. Resistance exercises at home were also recommended 5–10 repetitions three times per week at a heart rate intensity that was prescribed by a health professional [24, 25].

## Recommendations regarding flexibility

Very few publications provided detail or recommendations regarding the inclusion or suitability of flexibility exercises. Of the eight publications that included detail on flexibility, recommendations ranged from arm stretching for those who were wheelchair bound [22] to trunk twists, lateral bends and standing elbow to knee for intermediate and high functioning patients [37]. Five publications recommended flexibility exercises, but no detail or examples were provided [26, 27, 31, 38, 39] (S3 Table). One publication specifically recommended balance and flexibility exercises for those at high risk of falls but no additional detail was provided [31]. The most detailed recommendations on flexibility were reported in the ACSM guidelines [30], which provided advice on duration and type of flexibility exercises (60 seconds per joint for static stretches and 3–6 second contraction followed by 10–30 seconds of contraction at 20–75% of maximum voluntary contraction). Ten minutes was described as an appropriate duration for flexibility exercises [30, 39] (Table 2).

## Recommendations regarding progression of exercise

Details regarding progression of exercise were scarce. A summary of these recommndations is shown in Table 2 with additional detail in S3 Table. Eight publications recommended the progression of aerobic exercise be commenced in a graduated manner [23–26, 30, 33, 34, 38], with one recommending progression be based on individual tolerance but guided by a qualified exercise professional [30]. One publication recommended no aerobic exercise in the first two

weeks of an exercise program and to focus on strength training [23]. The majority of publications made no recommendations regarding progression [22, 27–29, 31, 32, 35–37, 39, 40].

## Safety recommendations

Most publications included safety recommendations for patients receiving dialysis treatment. The most common recommendations was that medical clearance was required before commencement [23–26, 30–32, 34–36, 38, 39] (Table 2); Most but not all guidance provided recommendations on absolute and relative contraindications to exercise [24, 26, 30, 31, 33, 35, 38, 39]. A number of papers detailed precautions for exercise (such as timing of exercise, monitoring using RPE and not heart rate, and reasons for cessation [23–26, 30, 31, 33, 35, 36, 38, 39] (S4 Table). Few publications gave specific guidance on safety precautions for those with an AVF [25, 30, 31, 35, 36, 38, 39]. Where included, the most common recommendation was to avoid placing weights on the fistula site (Table 2). For people undertaking PD guidance was provided to avoid increasing intra-abdominal pressure and commentary given on fill volume. [22, 24, 26, 30–32, 35–39]. No single publication reported on all of these safety aspects. Each of the safety recommendations are discssed in more detail below.

## Recommendations regarding medical clearance

Table 2 shows that medical clearance prior to participation in a structured exercise program was frequently recommended [23–26, 30–32, 34–36, 38, 39]. The majority indicated physical assessment was warranted prior to commencement [24, 25, 30–36, 38, 39]. An emphasis on cardiovascular exercise testing prior to commencement of exercise [24, 25, 31, 32, 34–36, 38, 39] and/or physical function was recommended [23, 34, 39]. The health professional responsible for providing clearance varied from a qualified clinical exercise specialist (specialized physical therapist); a physician (such sports medicine, cardiologist, diabetologist, nephrologist) [23–25, 32], and one suggested a 'healthcare professional' [30] (Table 2). Seven publications recommended medical clearance without specifying the healthcare provider responsible [31, 34–36, 38, 39] (S4 Table).

## Contraindications to exercise

A large number of conditions were described as contraindications to exercise (Table 2). This included those with an unstable cardiovascular status [30, 31, 33, 38, 39]. Only one publication identified cardiac insufficiency rated at New York Heart Association (NYHA) II or greater and valve disease greater than NYHA III [38] as a contraindication (S3 Table). Uncontrolled arrhythmias [30, 31, 33, 38, 39] and hypertension [30, 33, 35, 38, 39] were also reported as contraindications. There was substantial incongruence regarding ceiling blood pressure parameters that precluded exercise. Reported ceiling levels included: >160/100 mmHg [33], >180/110 mmHg [39], and systolic blood pressure (SBP) >200 mmHg or diastolic blood pressure (DBP) >110 mmHg [30, 35] (S3 Table). Recent myocardial infarction [30, 33, 35] and changes to electrocardiogram [31, 35] were also recommended as exclusion criteria. The definition of 'recent' infarct varied from 2 days [30] to 8 weeks [33] while no definition was provided in two publications for 'recent' changes in electrocardiogram [31, 35].

A distinct lack of guidance was apparent regarding the suitability to exercise for those with comorbidities. This included the mention of but no guidance for exercise in people receiving dialysis who had unstable angina [30, 33, 38], heart failure [30, 33, 39], severe and symptomatic aortic stenosis [30, 33, 39], pulmonary embolism/ infarction [30, 33], severe pulmonary hypertension [33, 39], deep vein thrombosis [30, 33], aortic / aneurysmal dissections [30, 39], or severe retinopathy [38]. Acute inflammation or infection was considered a preclusion to

exercise in two publications including pericarditis [30, 33], myocarditis [30, 33], and endocarditis [30]. Only two publications [26, 33] indicated that the presence of fever was a contraindication to exercise and only one provided temperature parameters [26].

Limited specific recommendations for dialysis patients with diabetes was identified. Of those that provided granular information (S4 Table), one suggested blood glucose >16.7 mmol/L (>300 mg/dL) with ketosis, or blood glucose <5.5 mmol/L (<99 mg/dL) warranted deferral from exercise [33]. Other specific advice suggested deferral from exercise if blood glucose levels were >13.9 mmol/L (250 mg/dL) with ketosis, or blood glucose <6.1 mmol/L (<110 mg/dL) [26]. Another publication suggested an ideal blood glucose for interdialytic exercise between 5.0–8.3 mmol/L (90–149 mg/dL) [23].

Electrolyte, fluid, and dialysis-related contraindications were also reported [25, 30, 31, 33, 35, 38]. Abnormal electrolyte levels were an indicator to defer exercise [30, 31, 35, 38], with hypo and hyperkalemia considered a contraindication criteria [31, 38]. In addition, one publication specified a temporary cessation of exercise if repeated hyperkalemia >6 mmol/L [38]. Significant peripheral edema was also reported as a contraindication [31, 33]. Those with an inter-dialytic weight gain of >4 kg was suggested by one publication as a preclusion from intra-dialytic exercise [31]. Hemodynamic stability during dialysis treatment was also considered important, and any patients considered unstable during dialysis were unsuitable for intra-dialytic exercise (IDE) [31]. For those undertaking PD, recommendations to defer exercise included peritonitis, catheter related infection and a non-functioning catheter [24].

### Recommendations regarding precautions for exercise

Table 2 summarises precautions obtained from 11 publications regarding exercise in dialysis patients [23–26, 30, 31, 33, 35, 36, 38, 39]. Additional information is shown in S4 Table.

### Precautions for dialysis patients

A high degree of variance existed regarding methods to monitor the intensity of exercising dialysis patients (S4 Table). The Borg RPE scale was recommended to monitor intensity, instead of heart rate [30, 35, 36, 39] with one publication providing the justification that RPE should be used over heart rate in patients on beta blockers due to a reduced maximal and submaximal exercise capacity [39]. It was noted that heart rate was not a reliable indicator to monitor exercise intensity in patients on hemodialysis [26, 30, 36]. The use of telemetry [25, 31] was recommended, with one recommending the use of telemetry in all home exercising patients [25], and the other recommending extensive monitoring in those prone to hypo/hyperkalemia [31]. One publication recommended the use of RPE, however the exercise prescription was based on heart rate [38]. Another publication recommended monitoring both heart rate and RPE throughout intradialytic exercise sessions [33].

No precautions regarding exercise in heat or humidity for interdialytic exercise were identified. Advice to warm up and cool down was provided, but this varied in the level of detail. Limited guidance on warm up and cool down was found other than to encourage the practice [35, 36, 38], while others recommended 3–4 minutes [26]; 5 minutes [33, 37]; 4–6 minutes [23]; or 5–10 minutes [24, 25].

Exercise termination criteria was addressed by most publications. These are summarised in Table 2 and additional detail in S4 Table. The most commonly cited reasons to stop exercise included shortness of breath [24, 30, 33, 39], angina [24, 30, 33, 39], dizziness/lightheadedness [24, 30, 33, 39], joint/muscle pain, nausea/vomiting, and discoloration of the face [24, 33]. If these signs or symptoms were encountered reducing exercise intensity [38] rather than termination of exercise was recommended. Concerns were expressed regarding hypotensive

episodes with post dialysis exercise [30], late IDE [35, 36], exercising on vasodilators [39], and exercising in hot or humid environments [39]. Only one publication included a significant drop in blood pressure as a termination criteria [30]. ACSM [30] specified a drop in SBP >10 mmHg with an increased workload and other signs of ischemia warranted the termination of exercise. Unlike most other publications, ACSM provided an extensive list of termination criteria including general and clinical criteria [30].

## Specific precautions for hemodialysis patients

For hemodialysis patients performing IDE, exercise was recommended within the first 2 hours of dialysis [30–32, 35, 36], suggesting this reduced risk of hypotension [35, 36], contrary to another paper which suggested late IDE could mitigate hypotension in certain individuals [30]. The ACSM advised avoiding participation immediately following hemodialysis treatment due to the increased risk of hypotension [30]. Exercise on non-dialysis days was recommended [25, 26, 30–33, 35, 36], and or avoiding post dialysis exercise [26, 30].

## Specific precautions for peritoneal dialysis patients

For those receiving PD treatment, caution was advised with exercises that increased intra-abdominal pressure [32, 38]; however, one reported that even though activities such as straining and weightlifting may increase intra-abdominal pressure, this effect may be of little clinical relevance [37] (S3 Table). Positions to avoid included extensive twisting of the torso, spinal flexion, pulling legs or knees towards the abdomen, abdominal static strength, lying prone or on the side without support [38], strong isometric resistance exercises [32], and intense and long contractions of the abdominal muscles [24]. One recommendation included monitoring the catheter site specifically for increased pressure with upright cycling [22]. Focus on maintaining a steady breathing pattern and avoiding the Valsalva maneuver was also recommended [24, 25, 38]. Disparities existed regarding fluid volume in the peritoneal cavity prior to exercise (Table 2). To reduce discomfort, exercising with an empty peritoneal cavity was recommended [30, 31]; or partially filled cavity [26, 38, 39]; and one indicated either empty or partially filled would reduce diaphragmatic discomfort [26]. Maintaining enough fluid 'to allow for floating of the catheter' may mitigate discomfort [39]. Paediatric recommendations included starting with fluid in the cavity and exercising empty if not tolerated well [35, 36]. No definition of 'partial fluid' nor intolerance was provided. One publication recommended patients exercise with a full cavity (up to 2L) and suggested that only patients with a history of leaks, hernias, or hypotension exercise with an empty cavity [24].

## Access site precautions for patients

Access site precautions were reported in many publications [22, 25, 30–32, 35–39] and are summarised in Table 2. Additional details are shown in S4 Table. For hemodialysis patients using an AVF as the access site, recommendations to avoid exercising the AVF arm during dialysis treatment were offered [30, 31, 39]. In contrast, four publications stated that the AVF arm was permitted to exercise [30, 35, 36, 39]; however, no guidance on how much weight could be held by the AVF arm was included. Three publications specified that weight could not be directly placed onto the AVF [30, 35, 36], while one reported on the need for a newly created AVF to heal prior to exercise [39]. Regarding AVF arm positioning, only one publication advised against maintaining the arm above the head for prolonged duration but no timeframe was specified [38]. While many publications reported caution was required at the dialysis access site, many were found to be lacking in detail and vague such as 'not over-exerting the fistula arm' [22, 25, 28, 29, 32–34, 37, 40].

## Access site precautions for peritoneal dialysis patients

Recommendations regarding swimming/water safety for patients receiving PD are summarised in Table 2 [22, 24, 35, 37]. Although some publications advised dressing the access site if swimming [22, 37], some disparities among these recommendations were evident (S4 Table). This included recommendations to clean the access site and change the dressing immediately after swimming [26, 37], dress the access site if swimming [26, 35], and securing the catheter with an additional stoma bag for water sports [24]. Two publications included advice against swimming in bodies of water with high level of 'germs' such as lakes and ponds [22, 26], and one specifically recommended swimming in chlorinated or seawater instead [26].

Table 3 provides a condensed overview of the guidance found in the publications retrieved and a rating of the quality of evidence and the strength of recommendations contained within these publications. Fifteen publications provided opinion level guidance, while four produced guidance produced by a systematic review of the evidence.

**Table 3. Overview of guidance retrieved on physical activity and exercise with a rating of the quality of evidence and the strength of recommendations contained within these publications.**

|  | Physical activity | Aerobic exercise | Resistance exercise | Flexibility exercise | Evidence level for recommendations # | Year published |
|---|---|---|---|---|---|---|
| American College of Sports Medicine [30] | ✓ | ✓ | ✓ | ✓ | 1B and 2C | 2021 |
| Chilean Society of Nephrology [28, 29] | ✓ | ✓ |  |  | Opinion | 2020 |
| European Federation of Sports Medicine Association [27] |  | ✓ | ✓ | ✓ | Opinion | 2015 |
| Exercise and Sports Science Australia [31] |  | ✓ | ✓ | ✓ | Opinion | 2013 |
| Fuhrmann and Krause [38] |  | ✓ |  | ✓ | Opinion | 2004 |
| Heiwe and Jacobson [40] | ✓ |  |  |  | 1A | 2011 |
| Isnard- Rouchon et al [37] |  | ✓ | ✓ | ✓ | Opinion | 2019 |
| Italian Society of Nephrology [32] |  | ✓ |  |  | Opinion | 2015 |
| KDOQI [34] | ✓ | ✓ |  |  | 1B and 1C | 2005 |
| Life Options Rehabilitation Advisory Council [26] | ✓ | ✓ | ✓ | ✓ | Opinion | 1995 |
| Patel et al [35] |  | ✓ | ✓ |  | Opinion | 2009 |
| Polish Society of Nephrology [24, 25] | ✓ | ✓ | ✓ |  | Opinion | 2019 |
| Raj et al [36] |  | ✓ | ✓ |  | Opinion | 2017 |
| Renal Foundation of Inigo Alvarez de Toledo [23] |  | ✓ | ✓ |  | Opinion | 2021 |
| Roshanravan et al [39] |  | ✓ | ✓ | ✓ | Opinion | 2017 |
| Spanish Society of Nephrology [22] |  | ✓ | ✓ | ✓ | Opinion | 2020 |
| UK Renal Association [33] |  | ✓ | ✓ |  | 1B and 2C | 2019 |

# GRADE criteria[1] was used to rate the quality of evidence. Level 1 is a strong recommendation; level 2 a weak recommendation; an A rating suggests high confidence that the true effect is similar to the estimated effect; a B rating suggests moderate confidence in the effect estimate: The true effect is likely to be close to the estimate of the effect, but there is a possibility that it is substantially different, a C rating indicates low confidence and the effect might be markedly different from the estimated effect, and a D suggests very little confidence in the effect estimate. The true effect is likely to be substantially different from the estimate of effect. OPINION statements may be based on expert judgement with or without evidence reviews and are intended to provide guidance to clinicians if the evidence base was low quality or insufficient in size to write a graded recommendation.

1 GRADE working group. GRADE Handbook. In: Schunemann H, Brozek J, Guyatt G, Oxman A (eds.). *Handbook for grading the quality of evidence and the strength of recommendations using the GRADE approach*. 2013. Available from https://gdt.gradepro.org/app/handbook/handbook.html

## Discussion

Few health professionals would dispute that physical activity and exercise provide physical, psychological, and social benefits for people receiving dialysis. This scoping review sought to synthesise current recommendations and identify areas that require further research or clarification. From the 19 publications retrieved, we were able to identify that there is no shortage of guidance on physical activity and exercise for patients who undertake dialysis. While there are some important variations in the level of detail, recommendations on aerobic exercise, progressive resistance training and flexibility were present in many publications. Several publications also provided detail on dialysis-specific topics such as protection of the AVF and exercise in those undertaking PD. New guidance released after the completion of the scoping review is now available for PD [41]. This is an important step forward for health professionals who wish to provide truly patient centred care and help dialysis patients achieve optimal life participation—outcomes indentified by patients as critically important to them [42, 43].

Importantly, the scoping review identified major inconsistencies between the available guidance, likely reflecting the low quality evidence base and dated nature of some publications. Inconsistencies were noted in exercise definitions as well as the advice on the frequency (sessions and minutes per week), intensity (assessment of, repetitions), type (aerobic, resistance, flexibility), timing (dialysis vs non-dialysis day; pre/intra/post-dialysis), progression (rate of increasing aerobic activity or weights over time), and safety aspects (medical clearance, contraindications, precautions) of exercise. Dialysis-specific advice also varied, including the presence of recommended volume of peritoneal dialysate during exercise, as well as recommendations regarding swimming and the impact of exercise on intra-abdominal pressure. Overall, the heterogeneity and discrepancies between the published guidance poses challenges for health professionals who wish to provide specific advice about physical activity and exercise to people undertaking dialysis.

The scoping review identified a scarcity of specific guidance about assessment of baseline fitness levels, suitability of exercise in the context of medical co-morbidities, and dialysis-specific exercise precautions (e.g. in patients prone to significant volume or electrolyte shifts, exercise considerations for those with an AVF). Future guidance should address these aspects so that health professionals can encourage appropriate participation in physical activity and exercise.

The terms 'physical activity' and 'exercise' were often used incorrectly and interchangeably across the publications. Though both are important, recommendations for physical activity and exercise should be distinctly different. Very few publications in this scoping review defined or provided specific details on the potential value of physical activity for dialysis patients. This is a missed opportunity to encourage the provision of advice. Physical activity is positively correlated with improvements in mental health, sleep quality and physical functioning scores and negatively correlated with overall mortality risk in people with kidney disease, [5, 44, 45]. A recent prospective observational study of hemodialysis patients reported an average of 3688 steps per day, with patients over 80 years recording only 1232 steps per day compared to younger patients with 4529 steps per day [46]. Without specific guidance and dialysis specific resource materials, the recommendations to achieve at least 150–300 minutes per week of moderate aerobic physical activity [21] are more difficult.

Other discrepancies with terminology were identified and related to descriptions of exercise intensity. Discrepancies such as this are not unique to nephrology, and position statements to standardise terminology used in exercise recommendations do exist [47]. Unfortunately, variations in definitions of exercise intensity among the dialysis population are not benign and may contribute to inaccurate and ineffective exercise prescription, and potentially cause patient

harm. For example, almost three quarters of publications in this review recommended 'moderate intensity' exercise. However descriptions of 'moderate intensity exercise' using the rating of Borg RPE ranged from 10–12 [23] / 11–13 [30, 31, 35, 36] / 12–13 [30] / 12–16 [26]/ or 11–16 [24, 25, 39]. These variations may be associated with the use of unauthorized 'adapted' versions of the original scale. The use of versions of the Borg RPE scale that differ from the original definitions [48] are not recommended. This is because deviations from the original scale are not consistent with the original psychological framework [49], and are associated with inconsistencies in achievement of desired exercise intensity [50].

This scoping review has also identified gaps in current guidelines regarding warm up and cool down for patients receiving dialysis. These elements are distinct components of an exercise training session that appear to be poorly described or not described at all. Given the greater risk of cardiovascular and musculoskeletal complications in the dialysis population [51], the need for guidance in this regard is important. In a healthy population, warm up serves to increase body temperature [52], improve muscle blood flow through vasodilation [53], increase nerve conduction [52], and reduce joint stiffness [52, 54]. The preparatory warm up phase may also reduce the likeliness of inducing cardiac ischemia from sudden effort [55, 56]. Immediately after exercise a recovery period occurs, which physiologically differs greatly from a warm-up or exercise training. Although the mechanisms of post exercise hypotension in a healthy individual varies between resistance and aerobic exercise [57, 58], the recovery period can be considered a 'vulnerable' phase where adequate precautions should be taken [59].

Given the high prevalence of cardiovascular disease in dialysis patients [60], the cardiac physiology of dialysis patients should also be explicitly considered in all future recommendations. Abrupt termination of exercise in patients with coronary heart disease results in a decrease in venous return, decrease in cardiac output, and a reduction in blood pressure [61]. Post exercise hypotension has also been documented in those in the general population with hypertension [62–64]. Additionally, those on calcium channel blockers, alpha blockers, and vasodilators may be particularly vulnerable to hypotension immediately following exercise [62, 65]. Cool downs, in the form of active recovery, immediately following exercise training can prevent post exercise hypotension [57, 62] and syncope [57]. Improving venous return and cardiac preload, via skeletal muscle contractions [57, 62], may prevent blood pooling in lower extremities [66]. It would appear reasonable to suggest that dialysis patients should perform an active cool down following exercise. Interestingly, the publications in this review deemed hypotension as an exercise precaution in dialysis patients, however recommendations for adequate cool downs were sparse. Given the lack of dialysis specific guidance for multimorbid patients, it would be reasonable to adopt guidance from organizations such as the ACSM whereby those with cardiovascular, pulmonary, and musculoskeletal comorbidities, are advised to warm up and cool down at a light intensity for 5–10 minutes. Patients suitable for an extended period up to 15 minutes, but not exceeding [19], includes those with heart rate delays with cardiovascular disease. Educating patients about the physiological rationale for warm up and cool down periods is also recommended.

There are several limitations to this review including the restriction of relevant articles to the English, Spanish, and Polish languages only. Documents in the grey literature may also have been missed despite a systematic search strategy. Several guidelines and articles included are also more than a decade old [26, 29, 34, 38] and may not reflect recent advancements in the field of exercise in kidney disease. Strengths of this work include the broad systematic nature of the search strategy utilising the major databases and supplementary search techniques thereby allowing a broad overview of the topic.

**Table 4. Summary of recommendations from scoping review findings.**

| |
|---|
| Reporting within future recommendations should: |
| • Define physical activity, aerobic and resistance exercise |
| • Include examples of physical activity |
| • Include details in the exercise prescription about the recommended timing for intradialytic exercise and interdialytic exercise |
| • Include explicit information regarding exercise precautions, contraindications and termination criteria |
| • Provide recommendations regarding precautions for the fistula or peritoneal dialysis catheter. |
| • Use the standard Borg Rating of Perceived Exertion (RPE) scale to describe intensity without modification |
| Exercise testing and prescription in future recommendations should: |
| • Recommend strength testing using 3–5 Repetition Maximum |
| • Incorporate information about the importance of warm up and cool down activities |
| • for at least 5–10 minutes (not exceeding 15 minutes) |
| • Describe moderate intensity using the Borg RPE scale |
| • Encourage exercise in a well-controlled environment during periods of extreme heat and humidity |
| Central clearinghouse and resource platform: |
| • Advocate for a central clearinghouse and information platform for exercise in dialysis patients. This could include the Global Renal Exercise Network (https://grexercise.kch.illinois.edu/) |

## Future directions

As a result of this scoping review, we propose several suggestions for future works that are summarised in Table 4. Firstly, specific guidance regarding physical activity should be constructed for people undertaking dialysis given the low physical fitness base. Secondly, guidelines for exercise should contain standardised reporting that includes details of the timing, duration, intensity (e.g. standardised RPE scores and 1-RM measures) frequency and progression of exercise. Specific aspects regarding safety and assessment of appropriateness to exercise should also be included, with explicit instructions on use of the AVF arm, and for people undertaking PD, advice regarding exercise with dialysate in the abdomen. A proposed checklist for authors who intend to publish guidelines for physical activity and exercise in people undertaking dialysis is provided in Table 5. This checklist can be used to improve clarity and standardise recommendations regarding details we have identified to be lacking or vague in the exercise prescription including exercise intensity; inclusion of specific information for patients with cardiorespiratory comorbidities, and details regarding warm up and cool down. The third recommendation that arises from this review is a suggestion for the development of a central clearinghouse for dialysis related exercise information. Given the challenges encountered locating relevant guidance documents for this review, organisations such as the Global Renal Exercise Network [67] could host relevant guidance and resources on their website and/or link health professionals to relevant documents. This may facilitate the dissemination and wider implementation of advice to people undertaking dialysis.

Recommendations for future research that arise from this review include the suggestion to test an adaption of the Borg scale in plain language for the dialysis population where cognitive impairment and low health literacy are highly prevalent [68, 69]. Research exploring staffing of exercise professionals among the multidisciplinary team working in dialysis centres would also be beneficial. Other recommendations that arise from this review include the need for future guidance on physical activity and exercise be directed towards patient-centred and patient important outcomes.

**Table 5.  Proposed checklist for reporting in physical activity and exercise guidelines for people undertaking dialysis.**

| |
|---|
| Physical Activity |
| • Definition of physical activity in plain language |
| • Examples of suggested activities |
| • Recommended duration of physical activity |
| • Recommended intensity with explanation in plain language using a Rating of Perceived Exertion (RPE) scale |
| Exercise |
| • Definition of exercise types (aerobic, resistance and flexibility and balance) |
| • Examples of suggested activities |
| • Information regarding pre- exercise screening / medical clearance with information on who should conduct the screening / assessment |
| • Information on precautions and absolute contraindications for each type of exercise |
| • Details of frequency, timing and duration of exercise including whether it is recommended to exercise pre / intradialytic or post dialysis |
| • Details regarding recommended intensity with explanation in plain language using a RPE scale |
| • Details regarding monitoring and progression of exercise |
| • Details regarding cessation including contraindications with specific mention of hypotension, electrolyte abnormalities, myocardial infarction and cardiac instability |
| • Specific institutions regarding safety and use of fistula arm |
| • Specific institutions regarding abdominal fluid and cleaning of peritoneal dialysis catheter site (if required) for those undertaking peritoneal dialysis |
| • Specific precautions for those with diabetes (if relevant) |
| • If home programs of exercise are suggested, then details in plain language are required for patients which include the above information |
| • Details regarding the types of exercise and duration of warm up and cool down periods (including patient types who may require additional cool down time) |

## Conclusions

Regular physical activity and exercise are beneficial for people receiving dialysis. While recommendations do exist, they differ substantially and lack guidance in many key areas. Collaborative multidisciplinary work that provides guidance in these areas may lead to increased participation in physical activity and exercise and facilitate better patient outcomes.

## Supporting information

**S1 Table. Grey literature search strategy.**
(DOCX)

**S2 Table. Recommendations for physical activity for people receiving dialysis.**
(DOCX)

**S3 Table. Recommendations for exercise for people receiving dialysis.**
(DOCX)

**S4 Table. Safety and other recommendations.**
(DOCX)

**S1 Checklist.**
(DOCX)

## Acknowledgments

Thank you to all members of the Global Renal Exercise Network who responded to our call for information and graciously shared resources.

## Author Contributions

**Conceptualization:** Kelly Lambert, Iwona Gabrys, Paul N. Bennett.

**Data curation:** Kelly Lambert, Iwona Gabrys.

**Formal analysis:** Kelly Lambert, Courtney J. Lightfoot, Dev K. Jegatheesan, Iwona Gabrys.

**Funding acquisition:** Kelly Lambert.

**Investigation:** Kelly Lambert, Iwona Gabrys.

**Methodology:** Kelly Lambert, Courtney J. Lightfoot, Paul N. Bennett.

**Project administration:** Kelly Lambert.

**Writing – original draft:** Kelly Lambert, Iwona Gabrys.

**Writing – review & editing:** Kelly Lambert, Courtney J. Lightfoot, Dev K. Jegatheesan, Iwona Gabrys, Paul N. Bennett.

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
