## [Decision Letter · Decision Letter 0]

22 Dec 2021

PONE-D-21-37604Physical activity and exercise recommendations for people receiving dialysis: A scoping reviewPLOS ONE

Dear Dr. Lambert,

Thank you for submitting your manuscript to PLOS ONE. After careful consideration, we feel that it has merit but does not fully meet PLOS ONE’s publication criteria as it currently stands. Therefore, we invite you to submit a revised version of the manuscript that addresses the points raised during the review process.

We look forward to receiving your revised manuscript.

Kind regards,

Pierre Delanaye

Academic Editor

PLOS ONE

Journal Requirements:

"Thank you to all members of the Global Renal Exercise Network who responded to our call for information and graciously shared resources. KL received funding from the Illawarra Health and Medical Research Institute to cover open access publication charges."

"KL. Funding was received from the Illawarra Health and Medical Research Institute for this work. https://www.ihmri.org.au/ 

3. We note that Figure 2 in your submission contain [map/satellite] images which may be copyrighted. All PLOS content is published under the Creative Commons Attribution License (CC BY 4.0), which means that the manuscript, images, and Supporting Information files will be freely available online, and any third party is permitted to access, download, copy, distribute, and use these materials in any way, even commercially, with proper attribution. For these reasons, we cannot publish previously copyrighted maps or satellite images created using proprietary data, such as Google software (Google Maps, Street View, and Earth). For more information, see our copyright guidelines: http://journals.plos.org/plosone/s/licenses-and-copyright.

Additional Editor Comments:

I agree with reviewers’ comments about tables 2 to 4. These tables are cumbersome, not easy to read, and just a repetition of the text. Table 5 must be kept.

I question the fact to consider at the same level recommendations from authors (ref 34 and 35 as an example) and recommendations from national societies.

I share the concerns of Reviewer 3 about the “recommendations on recommendations”. The authors should only focus on the clear result of their literature review, i.e. the heterogeneity in the recommendations and the need for further researches in the field. All points that could be considered as “personal opinion” (even if of interest) should be deleted or labeled as “opinion of the authors”. I also agree on the importance of describing the “level of evidence” in recommendations when available.

Reviewers' comments:

Reviewer's Responses to Questions

**Comments to the Author**

1. Is the manuscript technically sound, and do the data support the conclusions?

Reviewer #1: Yes

Reviewer #2: Yes

Reviewer #3: Partly

2. Has the statistical analysis been performed appropriately and rigorously? 

Reviewer #1: Yes

Reviewer #2: N/A

Reviewer #3: N/A

3. Have the authors made all data underlying the findings in their manuscript fully available?

Reviewer #1: Yes

Reviewer #2: Yes

Reviewer #3: Yes

4. Is the manuscript presented in an intelligible fashion and written in standard English?

Reviewer #1: Yes

Reviewer #2: Yes

Reviewer #3: Yes

5. Review Comments to the Author

Reviewer #1: This is a well written scoping review about the published recommendations on physical activity and exercise in patients benefiting from dialysis.

General comments:

- In the introduction, please detail the benefits that are expected from exercises and physical activities in this population. Please also detail a bit more their adverse effects.

- Some of the included publications were published more than 15 years ago. I would suggest limiting the selection to at least the 10 past years only, if authors want their work to reflect the current state of art.

- Tables 2-3-4 are very long and somehow redundant with the text. They should rather be inserted as supplemental material. On the contrary, it could be interesting to add a table summarizing the common recommendations among the included publications. Such table could also include the major ranges of differences, as explain in the results. It could help the reader to get an overview about what should be done with the patients, even if there are lots of gaps and discrepancies in the included publications.

Minor comments:

- L459: please correct the sentence: “ … with the USE (?) of …”

- L506: please specify that these recommendations you mention are related to future works

Reviewer #2: Thank you for giving me an opportunity for reviewing interesting paper.

The authors explore the recommendations for physical activity and exercise for patients undertaking dialysis in this scoping review paper. As a results of screening, 19 publications were eligible for inclusion, and among these, 13 publications were published on behalf of professional associations or foundations such as ACSM. While these publications provided each recommendations of physical activity and exercise for dialysis patients, they differ substantially and lack guidance in many key areas (e.g. exercise of timing, duration and intensity). From these results, authors summarize the recommendations which should be reported in future, and make the checklist for reporting in physical activity and exercise guidelines for dialysis patients.

The methods of this review are appropriate, the results are clearly presented and the conclusion are hardly controversial. This scoping review paper provides an important contribution to the field of physical exercise and activity in dialysis patients, I sure would like to see it printed soon.

Reviewer #3: The paper is a review of recommendations on physical activity and exercise for people receiving dialysis. Although literature has been widely studied, the paper is a list of existing recommendations and it does not bring much except in concluding that recommendations are very different one from another. A more important question to be answered would be: why do these recommendations vary so much? Which studies are they based on?

To finish, the authors provide interesting advices and a checklist for future recommendations. These “recommendations for the recommendations” are common sense but they do not include any reference and are mainly the author’s opinion. There again, a more important question to be raised could be: which trials are lacking to improve the grade recommendation and to better define which physical activity and exercise is beneficial to the patients.

Major Comments

The level of evidence and grade of recommendation should be provided for each recommendation when available.

Table 2 does not bring much. It is large and not easy to read and could be replaced by text.

Paragraph on safety recommendations should be re-written and include the safety recommendations. To only describe the number of papers providing such data is not very interesting...

The authors should define in the introduction or materials (rather than in the discussion) the terms “physical activity” and “exercise” and explain the difference.

Minor comments:

None

6. PLOS authors have the option to publish the peer review history of their article (what does this mean?). If published, this will include your full peer review and any attached files.

Reviewer #1: No

Reviewer #2: No

Reviewer #3: No

---

## [Author Response · Author response to Decision Letter 0]

16 Mar 2022

Response to reviewers

22/1/2022 

Dear Editor, 

We are pleased to submit the revised article for consideration for publication in your journal. Thank you to both reviewers and the editor for the useful feedback and very fast turnaround during the end of year period. 

We have made a number of changes in response to reviewer comments / suggestions and these are shown in track changes or red colour in the revised version.

In response to the query about funding, as directed we would like to include the proposed amended statement for funding: Funding was received from the Illawarra Health and Medical Research Institute for this work. The funders had no role in study design, data collection and analysis, decision to publish, or preparation of the manuscript” 

Yours Sincerely, 

Dr Kelly Lambert 

 

Journal requirements

We note that Figure 2 in your submission contain [map/satellite] images which may be copyrighted. All PLOS content is published under the Creative Commons Attribution License (CC BY 4.0), which means that the manuscript, images, and Supporting Information files will be freely available online, and any third party is permitted to access, download, copy, distribute, and use these materials in any way, even commercially, with proper attribution.

Response: We have now removed this Figure from the manuscript

 

Additional Editors Comments

I agree with reviewers’ comments about tables 2 to 4. These tables are cumbersome, not easy to read, and just a repetition of the text. Table 5 must be kept.

Tables 2-4 have been changed to Supplementary Tables based on reviewer feedback. We feel some readers may be interested in the granular detail provided so have elected to keep as supplementary material rather than remove entirely or reformat.

Previous Table 5 has been retained and is reformatted to include the evidence level of the recommendations retrieved and is now labelled Table 3 

A new table has been included which summarises the recommendations (a ‘super’ summary). This is labelled as Table 2 in the revised version and we hope this provides more clarity and improves the reader experience compared to the original version. 

I question the fact to consider at the same level recommendations from authors (ref 34 and 35 as an example) and recommendations from national societies.

We acknowledge the challenge of the reviewers when attempting to discern the recommendations described in the results section. To alleviate this, we have included a column in Table 3 which provides a rating of the quality of evidence and the strength of recommendations contained within the publications retrieved. In addition, in the new Table 2, we have bolded recommendations retrieved via a systematic review process so as to highlight recommendations based on opinion only. We hope this is helpful for readers. 

I share the concerns of Reviewer 3 about the “recommendations on recommendations”. The authors should only focus on the clear result of their literature review, i.e. the heterogeneity in the recommendations and the need for further researches in the field. 

We have changed the subheading in the discussion to future directions. This is to clarify that we are not making recommendations on recommendations, but rather, are making suggestions for future work that have been identified from this review

All points that could be considered as “personal opinion” (even if of interest) should be deleted or labeled as “opinion of the authors”. 

To help address this point, we have highlighted in the new Table 2, recommendations that were retrieved via a systematic review process (as opposed to opinion only). We hope this addresses the reviewer’s concern. 

I also agree on the importance of describing the “level of evidence” in recommendations when available.

The strength of the recommendations retrieved from the 19 publications is summarised in Table 3 and also now described in the manuscript (lines 415-418, 435-437):

Table 3 provides a condensed overview of the guidance found in the publications retrieved and a rating of the quality of evidence and the strength of recommendations contained within these publications. Fifteen publications provided opinion level guidance, while four produced guidance produced by a systematic review of the evidence.

Reviewer comments

Reviewer #1: 

- In the introduction, please detail the benefits that are expected from exercises and physical activities in this population. Please also detail a bit more their adverse effects.

Additional wording has now been included. “Remaining physically active is important to patients undertaking dialysis [6-9]. Exercise also produces many benefits including improved physical function [3], muscle mass and strength[10]. However, there are numerous barriers to undertaking regular physical activity…”

- Some of the included publications were published more than 15 years ago. I would suggest limiting the selection to at least the 10 past years only, if authors want their work to reflect the current state of art.

Thank you for the observation. We have elected to keep all eligible publications as these may still be used in clinical practice despite their vintage. We have however included reference to the nature of the age of these older publications in the results section lines 158-159. We have also updated new Table 3 with the year of publication.

- Tables 2-3-4 are very long and somehow redundant with the text. They should rather be inserted as supplemental material. On the contrary, it could be interesting to add a table summarizing the common recommendations among the included publications. Such table could also include the major ranges of differences, as explain in the results. It could help the reader to get an overview about what should be done with the patients, even if there are lots of gaps and discrepancies in the included publications.

Thank you for the suggestion / feedback. We have elected to move the table 2-4 to supplementary information as we believe some readers may be interested in the detail provided. We have also summarised this information into one table (new Table 2). We hope this provides clarity and a concise overview. 

- L459: please correct the sentence: “ … with the USE (?) of …”

Thank you – this is now corrected 

- L506: please specify that these recommendations you mention are related to future works

Thank you – we have now amended

Reviewer #2: no comments to address. Thank you for the feedback

Reviewer #3: The paper is a review of recommendations on physical activity and exercise for people receiving dialysis. Although literature has been widely studied, the paper is a list of existing recommendations and it does not bring much except in concluding that recommendations are very different one from another. A more important question to be answered would be: why do these recommendations vary so much? Which studies are they based on?

To address this concern, we have undertaken additional synthesis and this is shown in Table 2. We believe this additional summary of what was previously contained in tables 2-4, is a more useful method to gauge the areas where future work can be undertaken to improve the quality of advice on exercise for patients undertaking dialysis. 

To finish, the authors provide interesting advices and a checklist for future recommendations. These “recommendations for the recommendations” are common sense but they do not include any reference and are mainly the author’s opinion. There again, a more important question to be raised could be: which trials are lacking to improve the grade recommendation and to better define which physical activity and exercise is beneficial to the patients.

We have altered this section of the manuscript to ‘Future recommendations’ . We believe this better reflects the intent of this paragraph and is more aligned with the intention of the scoping review. In addition, we have now highlighted which of the guidance retrieved is based on systematic review evidence. While some sentences in this section may be common sense, we believe this is the first time, all previous guidance has been synthesised in such a manner.

The level of evidence and grade of recommendation should be provided for each recommendation when available.

The level of evidence has now been included in the new Table 3. Reference to the level of evidence is also included in the results lines 415-418 and discussion line 435-437. 

Table 3 provides a condensed overview of the guidance found in the publications retrieved and a rating of the quality of evidence and the strength of recommendations contained within these publications. Fifteen publications provided opinion level guidance, while four produced guidance produced by a systematic review of the evidence.

Table 2 does not bring much. It is large and not easy to read and could be replaced by text.

We have elected to move the table 2-4 to supplementary information as we believe some readers may be interested in the detail provided. As described above, we have summarised information from the original tables 2-4 into one table (new Table 2). We hope this provides clarity and a concise overview.

Paragraph on safety recommendations should be re-written and include the safety recommendations. To only describe the number of papers providing such data is not very interesting.

Thank you – this has been amended in the new Table 2 to be more informative and useful for readers.

The authors should define in the introduction or materials (rather than in the discussion) the terms “physical activity” and “exercise” and explain the difference.

These definitions have been moved from the discussion and included in the methods (line 134-137). Thank you for the suggestion.

---

## [Decision Letter · Decision Letter 1]

28 Mar 2022

PONE-D-21-37604R1Physical activity and exercise recommendations for people receiving dialysis: A scoping reviewPLOS ONE

Dear Dr. Lambert,

Thank you for submitting your manuscript to PLOS ONE. After careful consideration, we feel that it has merit but does not fully meet PLOS ONE’s publication criteria as it currently stands. Therefore, we invite you to submit a revised version of the manuscript that addresses the points raised during the review process.

We look forward to receiving your revised manuscript.

Kind regards,

Pierre Delanaye

Academic Editor

PLOS ONE

Journal Requirements:

Additional Editor Comments (if provided):

The article has been largely improved, and has the potential to be published in PlosOne. Some comments by Reviewer 3 need to be addressed.

Reviewers' comments:

Reviewer's Responses to Questions

**Comments to the Author**

1. If the authors have adequately addressed your comments raised in a previous round of review and you feel that this manuscript is now acceptable for publication, you may indicate that here to bypass the “Comments to the Author” section, enter your conflict of interest statement in the “Confidential to Editor” section, and submit your "Accept" recommendation.

Reviewer #1: All comments have been addressed

Reviewer #3: (No Response)

2. Is the manuscript technically sound, and do the data support the conclusions?

Reviewer #1: Yes

Reviewer #3: Yes

3. Has the statistical analysis been performed appropriately and rigorously? 

Reviewer #1: N/A

Reviewer #3: N/A

4. Have the authors made all data underlying the findings in their manuscript fully available?

Reviewer #1: Yes

Reviewer #3: Yes

5. Is the manuscript presented in an intelligible fashion and written in standard English?

Reviewer #1: Yes

Reviewer #3: Yes

6. Review Comments to the Author

Reviewer #1: All my comments have been addressed.

Reviewer #3: The revised version of physical activity and exercise recommendations for patients receiving dialysis has been very much improved compared with the first version. The simplified Table 2 is very practical and concrete and I would like to thank the authors for the work done. The authors also have improved the understanding of grades of recommendations. Nevertheless, some additional modifications and clarifications are needed.

Major comments

I very much appreciated the paragraphs on the “recommendations regarding precautions for exercise”. To better understand which patient (dialysis, PD and HD) is concerned by the precautions, maybe this paragraph could be separated in 3 subtitles:

- Common risks for dialysis patients (potassium, warming and cooling, monitoring intensity...)

- Specific risks for hemodialysis patients (avoid exercise after HD session, AVF related risks etc...)

- Specific risks for PD patients (intrabdominal pressure, access site precautions...)

Also, the risks of exercising in hemodialysis and peritoneal dialysis patients are not explained:

- Line 368: Could the authors be more explicit on the risk of increasing intra-abdominal pressure: Is it for dialysate reabsorption? For hernias?

- Line 401: Access Site Precautions for Patients. What could be the risk if weight is directly placed onto the AVF? Does it increase the risk of bleeding from fistula?

Table 2: Contraindications: The list of contra-indications is not clear.

- For example, “high BP >160/100mm/Hg” is proposed as a contra-indication, but is it chronic hypertension or the blood pressure at the moment of the exercise?

- “Not stable on HD” : what does this mean? Is it intradialytic hypotensive episodes?

- “uncontrolled HT >180/110” : what does HT mean? Is it hypertension? Then is it 160/110 or 180/110? This point is clear in the text (lines 303-305), but not in the Table.

This section could be simplified, maybe by sorting contra-indications.

Minor Comments

Table 2: Precautions to Exercise:

- “Avoid exercise in hot environment s including saunas” Please correct the typo.

- It is not clear to me whether blood pressure must be measured before exercise sessions and if it might be a contra-indication.

Table 2: Legend:

- “Bolded statements are based on higher quality evidence rather than opinion only”: please be more precise on the difference of evidence quality between bolded and non bolded statements. Is it the number of papers with the statement or a grade of level evidence (then precise)?

Line 277 “Where included, the most common recommmedation was avoid placing weights on the fistula site (Table 2).” Please correct typo error

Line 279: “For people undertaking PD guidance was provided to avoid increasing intra-abdoinal pressure and commentary given on fill volume).” Please correct typo error

7. PLOS authors have the option to publish the peer review history of their article (what does this mean?). If published, this will include your full peer review and any attached files.

Reviewer #1: No

Reviewer #3: No

---

## [Author Response · Author response to Decision Letter 1]

5 Apr 2022

Dear Editor, 

We are pleased to submit the revised article for consideration for publication in your journal. Thank you to both reviewers for the useful feedback. 

We have made a number of changes in response to reviewer comments / suggestions and these are shown in yellow in the revised version.

Yours Sincerely, 

Dr Kelly Lambert 

 

Reviewer 3 comments

1. I very much appreciated the paragraphs on the “recommendations regarding precautions for exercise”. To better understand which patient (dialysis, PD and HD) is concerned by the precautions, maybe this paragraph could be separated in 3 subtitles:

- Common risks for dialysis patients (potassium, warming and cooling, monitoring intensity...)

- Specific risks for hemodialysis patients (avoid exercise after HD session, AVF related risks etc...)

- Specific risks for PD patients (intrabdominal pressure, access site precautions...)

Thank you for this helpful suggestion. The changes suggested have been included and the subheadings are shown in yellow in the revised version.

2. Also, the risks of exercising in hemodialysis and peritoneal dialysis patients are not explained: for example

- Line 368: Could the authors be more explicit on the risk of increasing intra-abdominal pressure: Is it for dialysate reabsorption? For hernias?

As this is a review, we were very careful in only presenting the recommendations that were documented in the reviewed manuscripts. The risk/s of increased intra-abdominal pressure was not explicitly detailed in the cited sources, potentially given limited evidence in this space. Whilst one could postulate that increasing intra-abdominal pressure could promote hernia formation and diaphragmatic splinting, we do not believe that including such comments is within the scope of this review. In addition we have included a further citation that stated “this effect may be of little clinical relevance”. 

- Line 401: Access Site Precautions for Patients. What could be the risk if weight is directly placed onto the AVF? Does it increase the risk of bleeding from fistula?

Not placing weights on AVF would be part of normal care and routine patient education - including no tight clothes, watches, avoidance of sharp objects etc. The risks of placing a weight on the AVF include increased risk of thrombosis, bleeding and potentially infection. As above, including these potential mechanisms appears to be outside the scope of this review. Common sense would suggest not placing weights o AVF and we have not added to this.

3. Table 2: Contraindications: The list of contra-indications is not clear.

- For example, “high BP >160/100mm/Hg” is proposed as a contra-indication, but is it chronic hypertension or the blood pressure at the moment of the exercise?

This is stating the BP at the time of exercise, similar to other contraindications. The word “Prevalent” has been added to help clarity. It now reads: “New within 3 months to hemodialysis, hypo or hyperglycaemia, high prevalent BP >160/100mm/Hg, SOB, peripheral oedema, suspected or recent pericarditis, myocarditis, dissecting aneurysm, DVT, not stable on HD (eg intradialytic hypotension),...”

- “Not stable on HD” : what does this mean? Is it intradialytic hypotensive episodes?

This refers to intradialytic hypotension. The word has been added to the recommendation. 

- “uncontrolled HT >180/110” : what does HT mean? Is it hypertension? Then is it 160/110 or 180/110? This point is clear in the text (lines 303-305), but not in the Table.

This section could be simplified, maybe by sorting contra-indications.

Thank you for pointing this abbreviation out. HT refers to hypertension and has been updated in the table accordingly. 

Minor Comments

1. Table 2: Precautions to Exercise: “Avoid exercise in hot environment s including saunas” Please correct the typo.

This has now been corrected

2. It is not clear to me whether blood pressure must be measured before exercise sessions and if it might be a contra-indication.

Hypertension and Intradialytic hypotension have been stated as potential contraindications. Blood pressure is measured frequently pre, during and post-dialysis so this is a part of standard care. 

3. Table 2: Legend:- “Bolded statements are based on higher quality evidence rather than opinion only”: please be more precise on the difference of evidence quality between bolded and non bolded statements. Is it the number of papers with the statement or a grade of level evidence (then precise)? 

We have now clarified this with the statement: Bolded statements are derived from recommendations using systematic reviews or randomised controlled trials (higher quality evidence) rather than opinion only.

4. Line 277 “Where included, the most common recommendation was avoid placing weights on the fistula site (Table 2).” Please correct typo error

This has now been corrected

5. Line 279: “For people undertaking PD guidance was provided to avoid increasing intra-abdoinal pressure and commentary given on fill volume).” Please correct typo error

This has now been corrected

---

## [Editor Report · Decision Letter 2]

6 Apr 2022

Physical activity and exercise recommendations for people receiving dialysis: A scoping review

PONE-D-21-37604R2

Dear Dr. Lambert,

We’re pleased to inform you that your manuscript has been judged scientifically suitable for publication and will be formally accepted for publication once it meets all outstanding technical requirements.

Kind regards,

Pierre Delanaye

Academic Editor

PLOS ONE
---

## [Editor Report · Acceptance letter]

8 Apr 2022

PONE-D-21-37604R2 

Physical activity and exercise recommendations for people receiving dialysis: A scoping review 

Dear Dr. Lambert:

I'm pleased to inform you that your manuscript has been deemed suitable for publication in PLOS ONE. Congratulations! Your manuscript is now with our production department. 

Kind regards, 

on behalf of

Professor Pierre Delanaye 

Academic Editor

PLOS ONE